Journal of Data-centric Machine Learning Research (2025)      Submitted 1/25; Revised 06/25; Published 06/25

# Synthetic Datasets for Machine Learning on Spatio-Temporal Graphs using PDEs

**Jost Arndt**[1]                                          JOST.ARNDT@HHI.FRAUNHOFER.DE

**Utku Isil**[1]                                            UTKU.ISIL@HHI.FRAUNHOFER.DE

**Michael Detzel**[1]                                  MICHAEL.DETZEL@HHI.FRAUNHOFER.DE

**Wojciech Samek**[1,2,3]                      WOJCIECH.SAMEK@HHI.FRAUNHOFER.DE

**Jackie Ma**[1]                                          JACKIE.MA@HHI.FRAUNHOFER.DE

[1] *Department of Artifical Intelligence, Fraunhofer Heinrich Hertz Institute, Berlin*
[2] *Department of Electrical Engineering and Computer Science, Technische Universität Berlin*
[3] *BIFOLD – Berlin Institute for the Foundations of Learning and Data*

**Reviewed on OpenReview:**  *https://openreview.net/forum?id=EguDBMechn*

**Editor:** Yi Liu

## Abstract

Many physical processes can be expressed through partial differential equations (PDEs). Real-world measurements of such processes are often collected at irregularly distributed points in space, which can be effectively represented as graphs; however, there are currently only a few existing datasets. Our work aims to make advancements in the field of PDE-modeling accessible to the temporal graph machine learning community, while addressing the data scarcity problem, by creating and utilizing datasets based on PDEs. In this work, we create and use synthetic datasets based on PDEs to support spatio-temporal graph modeling in machine learning for different applications. More precisely, we showcase three equations to model different types of disasters and hazards in the fields of epidemiology, atmospheric particles, and tsunami waves. Further, we show how such created datasets can be used by benchmarking several machine learning models on the epidemiological dataset. Additionally, we show how pre-training on this dataset can improve model performance on real-world epidemiological data. The presented methods enable others to create datasets and benchmarks customized to individual requirements. The source code for our methodology and the three created datasets can be found on github.com/Jostarndt/Synthetic_Datasets_for_Temporal_Graphs.

**Keywords:**   Graph, Spatio-Temporal, PDE, Epidemiology

## 1 Introduction

Partial differential equations (PDEs) describe numerous scientific processes, and many of these processes are of spatio-temporal nature, for instance fluid dynamics, chemical diffusion, financial mathematics, climate modeling, biomechanics, or seismology. Over the past centuries, considerable research efforts have been dedicated to advancing mathematical modeling with PDEs. Although there exists a significant amount of work on PDEs and their solutions (Evans, 2022), the practical implementation of a PDE solver remains a demanding task. Prominent ML research focuses on the study of PDEs itself, such as Raissi et al. (2019); Li et al. (2021). To support this research, some public datasets for machine learning (ML) on PDE-generate data already exists (Takamoto et al., 2022), although they have some limitations that we will address later.

Graphs are a powerful tool for describing complex geometric structures in real-world scenarios, such as surfaces, volumes, or sensor networks. Recently, there have been notable developments in specifically spatio-temporal graph modeling within the field of machine learning (Chen and Eldardiry, 2024). One big obstacle for benchmarking, comparing, and for further developments, is the availability of spatio-temporal data. Only few datasets for spatio-temporal graph machine learning can be found in literature, such as traffic data from the *California PeMS system* (Li et al., 2018; Yu et al., 2018; Cini et al., 2023) in different settings, or COVID-19 data Gao and Ribeiro (2022). For some benchmarking applications Huang et al. (2023); Gastinger et al. (2024) offer different datasets of interest, Rozemberczki et al. (2021) offers access to some datasets with varying size, scope and quality. Although the existent datasets are valuable for specific tasks, they have limitations in quality (high noise), scope (few samples), application (topic), accessibility (unpublished), and adaptability (high individual pre-processing effort).

Thus, we identify a lack of larger high-quality datasets for different tasks. These limitations hinder the development and comprehensive benchmarking of spatio-temporal graph architectures for specific applications and diverse tasks. Therefore, large and more diverse high-quality datasets that are freely accessible are crucial.

To overcome these limitations, synthetic data is a promising ressource. The utilization of synthetic data allows the fast creation and adaption of datasets, while avoiding barriers regarding privacy or data availability. Furthermore, synthetic data is less error-prone regarding measurement flaws and easier to control, as all involved dynamics are known in contrast to real-world data. Some examples of (partially) synthetic datasets that find application in machine learning are extensively used weather datasets (Muñoz Sabater et al., 2021), turbulence data (Li et al., 2008), PDE data (Takamoto et al., 2022) or ground motion data (Lehmann et al., 2024).

This paper builds on the idea to utilize synthetic data and makes the significant developments from past PDE research accessible to the spatio-temporal graph learning community. We outline a method for generating synthetic datasets based on PDEs, and provide three spatio-temporal datasets ready to use, alongside with code. To this end, we solve three exemplary PDEs related to different types of disaster, each describing the movement of entities over space and time. We solve these PDEs on an irregular domain, and evaluate them on irregularly distributed points on this domain, as this is the realistic scenario for the presented applications. Subsequently, we form time-dependent graphs from the obtained

values in combination with the underlying spatial structure, to which we grant researchers direct access. To demonstrate the utility of our synthetic data, we lastly present a benchmarking of prominent spatio-temporal prediction models on the epidemiological data. We further underline the impact and importance of our synthetic data by successfully transferring knowledge onto real-world data.

Our key **contributions** can be summarized as follows:

- This paper presents three datasets, which constitute the first collection of PDE-based spatio-temporal datasets specifically designed for graphs.

- Through the detailed publication of code along with our method, we strongly encourage other practitioners to adapt our method. To ensure flexibility regarding sampling rate, domain (region), graph, dynamic, boundary condition, and PDE, we utilize the finite element method (FEM). We facilitate other temporal graph ML researchers to integrate PDE knowledge into their specific field and application.

- To our knowledge, our work provides the first numerical solution of this (or similar) epidemiological PDE, connecting the well-known SIR-ODE with spatial diffusion.

- To our knowledge, the epidemiological dataset constitutes the largest high-quality public epidemiological spatio-temporal graph dataset of this kind. This dataset allows us to offer new reproducible benchmarking results and give other practitioners the opportunity to develop enhanced models on our publicly available high-quality datasets, generated through a controlled setting.

- First pre-training of epidemiological spatio-temporal graph models on synthetic data and transfer-learning onto real-world data. This marks a significant step towards foundational modeling. Performance increase of up to 45% in our experimental setup.

Our work differs from the aforementioned (Takamoto et al., 2022) not only through the selected PDEs, but also by the fundamental technique, the FEM and adaptability. Specifically, the FEM allows for complex domains with complex boundary conditions. Also, our overall outcome is a temporal graph, not a grid. Further, we notably emphasize the interchangeability of the (complex) domain, dynamics, or underlying equation throughout our published code and method. Our work facilitates the adaption of our code for individual needs and applications.

## 2 Temporal PDEs to Simulate Spatio-Temporal Movement

To simulate propagations through space and time through three exemplary PDEs, we first generally outline the numerical approximation of the solution. In Section 3, we describe the generations of the datasets more explicitly.

**Overview of Our PDEs** The **first equation** we employ is inspired by the structure of real-world epidemiological data (e.g. COVID-19 incidences), which suggests the use of spatio-temporal graph machine learning (Nguyen et al., 2023; Gao and Ribeiro, 2022). We create a comparably large graph dataset containing long records of infectious data with different epidemiological settings but consistent measurements over time, absence of noise,

and unconstrained accessibility to all determining information. This or any similar epidemiological PDE, based on the SIR-ODE (Kermack and McKendrick, 1991), has never been solved numerically. Especially, there are no accessible simulations for other practitioners. Furthermore, we want to enable epidemiological researchers, to use individual spatial domains (e.g. country) by adapting our published code to specifically train and test models on other geometries. Additionally, a broad collection of similar PDEs with slight adaptations can easily extend the modeling by features such as vaccination status (Schlickeiser and Kröger, 2021), exposed (He et al., 2020), or disease vectors (Collins and Duffy, 2022). Naturally, similar adaptations can also be applied to the following two equations.

The **second equation**, an advection-diffusion (or convection-diffusion) equation, models the movement of particles (e.g. dust, nuclear fallout, smoke) or other quantities in the atmosphere. The arrangement as a spatio-temporal graph as the underlying datatype is canonical, as real-life measurements of those quantities with a sensor network likewise have an irregular geometry. Besides a further benchmarking of temporal graphs, the creation of such a dataset could be used to pre-train either risk-prediction models or to pre-train or test specific parts of ML-based weather models such as ClimODE (Verma et al., 2024).

As a **third PDE**, we choose the wave equation, which is very common in physics and describes, for example, water, sound, or electromagnetism. In our case, we model a tsunami wave approaching some coastline, where an interesting task could be to predict the next few steps of the wave.

**General setting**  We solve the PDEs on (different) 2D-domains $\Omega \subset \mathbb{R}^2$ for the three proposed scenarios. The PDEs are time-dependent on an interval $[0, T]$, and their solutions $\nu$ are of dimension $d \in \{1, 2\}$, depending on the equation. Generally, our PDEs have the form $\frac{\partial \nu}{\partial t} = F(\nu, \nabla \nu, \Delta \nu)$ with some additional boundary conditions. $F$ is some (general) functional which will be specified later, $\Delta$ the Laplace operator and $\nabla$ the Gradient, both only along the spatial dimensions.

The solution $\nu : [0, T] \times \Omega \to \mathbb{R}^d$ will be evaluated on a set of points within the domain $\Omega$. These points mimic geographical regions and their administrative units, or any unevenly distributed network of sensors. We equip this set of points with adjacencies and inverse distances to create a spatio-temporal graph dataset.

Our method is applicable to an even broader class of PDEs than those described here (e.g., for $d > 2$). It not only facilitates the exchange and extension of the domain or parameters, but can also be seamlessly extended to PDEs for which the FEM is effective, such as elasticity-, (Navier-)Stokes-, Euler-, Schrödinger-, and Burgers equations, for which numerous implementations are available.

**Numerical solution sketch**  In the following paragraphs, we describe the numerical approximation of different time-dependent PDEs. This part can be skipped by a reader looking for a spatio-temporal graph dataset but does not want to bother with the details of its generation. However, to understand and adapt the published code and the data generation process, a short sketch of the methods is inevitable to us and important to a reader looking to expand our set of equations and parameters. Note, that although the outlined methodology can be applied to countless processes and PDEs, the complexity of the synthesized data is limited by the amount and complexity of given dynamics.

We solve the equations iteratively in the time domain and solve the occurring PDE for each time-step with the FEM, this order is called Rothe method. We therefore first discretize alongside the temporal dimension with an Euler scheme, i.e. approximate the time-derivative with a difference quotient. We use the Crank-Nicolson method, which is a mixture of the implicit and explicit Euler scheme for time-stepping, which is known for good stability and accuracy with still only moderate implementation complexity. With $\theta = 0.5$ (Crank-Nicolson), a small temporal Euler stepsize $h$, and $\nu = \nu(x,t)$ and $\nu_+ = \nu(x, t + h)$ the approximation has the form

$$\frac{\nu_+ - \nu}{h} = \theta F(\nu, \nabla\nu, \Delta\nu) + (1 - \theta)F(\nu_+, \nabla\nu_+, \Delta\nu_+). \tag{1}$$

Since for the timestep $t$ the solution $\nu$ is known, we thereby obtain a PDE that is not time-dependent anymore. We solve this PDE for $\nu^+$ with the FEM, which is the most flexible, reliable, and therefore standard numerical approach on PDEs. We only give a sketch of the FEM here, as details can be found in numerical textbooks for PDEs. The FEM discretizes the domain $\Omega$ into a mesh, on which a set of test functions $\phi$ as well as trial functions $\tilde{\nu}$ (approximation of the solution $\nu$) are defined. Throughout this work, we use first-order (linear) Lagrange Elements as test and trial functions. To obtain the so-called weak formulation, we rearrange the PDE into an implicit and an explicit part, then multiply it from the left with the test functions and integrate over the full domain. This forms an inner product, which allows an underlying Sobolev space to have the structure of a Hilbert space. We will not dive deeper into this, but use the notation of an inner product $\langle\phi, \nu\rangle = \int_\Omega \phi\nu \, dx$ for a better readability.

Replacing additionally $\nu, \nu_+$ with the corresponding trial functions $\tilde{\nu}, \tilde{\nu_+}$, we obtain the formulation:

$$\langle\phi, \tilde{\nu}_+\rangle - h(1 - \theta)\langle\phi, F(\tilde{\nu}_+, \nabla\tilde{\nu}_+, \Delta\tilde{\nu}_+)\rangle = \langle\phi, \tilde{\nu}\rangle + h\theta\langle\phi, F(\tilde{\nu}, \nabla\tilde{\nu}, \Delta\tilde{\nu})\rangle.$$

The left-hand-side (implicit side) can be written as $A(\tilde{\nu}_+)$, the right hand (explicit side) as $b$. We then seek to solve $A(\tilde{\nu}_+) = b$ for (the parameters of) $\tilde{\nu}_+$. Note, that technically $\phi$ is a vector of test functions, but we wanted to keep the notation simple.

One often employed identity for compact $\phi$ is $\langle\phi, \Delta\nu\rangle = -\langle\nabla\phi, \nabla\nu\rangle$ due to integration by parts, which we apply when Dirichlet or zero-flow Neumann boundary values are imposed. Note that we omit plenty of information on which space this inner product is defined and details of such transformations for the sake of simplicity, but again refer to any textbook covering the FEM.

Further, we want to mention the most prominent downside of the FEM, which is its poor scaling behavior for high-dimensional problems: the number of support points usually increases exponentially with the dimensionality of the domain (curse of dimensionality). Additionally the matrix $A$ grows quadratically (but is extremely sparse) with the amount of support points. However, fine meshes are generally advantageous for obtaining accurate solutions. To address these computational challenges, various techniques have been developed, including adaptive mesh refinement, algorithms for sparse matrices, matrix-free solvers, and specialized computational tricks for individual PDEs. While these considerations are essential, an in-depth discussion of them falls outside the scope of our paper, as they are thoroughly addressed in textbooks and continue to be an active area of research.

## 2.1 SI-Diffusion Equation

To model the spatio-temporal spread of infectious diseases, we consider the following PDE from Murray (2003)

$$
\begin{aligned}
\frac{\partial S}{\partial t} &= -rIS + D\Delta S, \\
\frac{\partial I}{\partial t} &= rIS - \alpha I + D\Delta I,
\end{aligned}
\tag{2}
$$

where $\Delta$ denotes the Laplace operator and the functions $S(x,t), I(x,t)$ describe the densities of the susceptible and infected population over space and time.

The functions $\alpha(x,t), r(x,t)$, and $D(x,t)$ in the equation represent dynamics rising from pathogens and the population. More precisely, $r$ describes the transmission rate of the disease, $\alpha$ describes the duration of the disease, and $D$ describes the speed of the diffusion, i.e. movement of the population. Note that by setting $D = 0$ one receives the underlying SI compartment ODE, while setting $r = \alpha = 0$ leads to two time-dependent heat equations. Many adaptions of the underlying SI compartment ODE (He et al., 2020; Collins and Duffy, 2022) can easily be integrated into our framework.

As boundary condition, we chose mixed boundary conditions: usually, we impose zero-flow Neumann boundary conditions. However, to start a wave of infections, we impose small positive Dirichlet boundary conditions for a small part of the boundary for a limited time. Since this PDE is a system of two equations, the weak formulation utilizes two test functions $\phi_1, \phi_2$ from the same set of functions. The left-hand-side of the weak formulation, named $A$ above, here takes the form

$$
\begin{aligned}
A(\tilde{S}_+, \tilde{I}_+) = \langle \phi_1, \tilde{S}_+ \rangle &+ (1-\theta)h_k\langle \phi_1, r\tilde{S}_+\tilde{I}_+ \rangle + (1-\theta)h_k\langle \nabla\phi_1, +D\nabla S_+ \rangle \\
&+ \langle \phi_2, \tilde{I}_+ \rangle - (1-\theta)h_k\langle \phi_2, r\tilde{S}_+\tilde{I}_+ - \alpha\tilde{I}_+ \rangle + (1-\theta)h_k\langle \nabla\phi_2, D\nabla\tilde{I}_+ \rangle
\end{aligned}
$$

while the right-hand-side $b$ is defined by the following

$$
\begin{aligned}
b = \langle \phi_1, \tilde{S} \rangle &- \theta h_k\langle \phi_1, r\tilde{S}\tilde{I} \rangle - \theta h_k\langle \nabla\phi_1, +D\nabla\tilde{S} \rangle \\
&+ \langle \phi_2, \tilde{I} \rangle + \theta h_k\langle \phi_2, r\tilde{S}\tilde{I} - \alpha\tilde{I} \rangle - \theta h_k\langle \nabla\phi_2, D\nabla\tilde{I} \rangle.
\end{aligned}
$$

Due to the product $\tilde{S}_+, \tilde{I}_+$, the equation $A(\tilde{S}_+, \tilde{I}_+) = b$ depends nonlinearly on (the parameters of) the trial functions $\tilde{S}_+, \tilde{I}_+$. To solve this equation for (the parameters of) $\tilde{S}_+, \tilde{I}_+$, the Newton method is a standard choice, which we used. The Newton method solves this system iteratively, i.e. starts with $\tilde{S}_{+,1}, \tilde{I}_{+,1}$ and calculates $\tilde{S}_{+,2}, \tilde{I}_{+,2}$ until a convergence criterion is reached. The updates $\delta_S, \delta_I$ to $\tilde{S}_{+,i}, \tilde{I}_{+,i}$ are calculated as the solution of the linear system $J_A(\tilde{S}_{+,i}, \tilde{I}_{+,i})\delta = -A(\tilde{S}_{+,i}, \tilde{I}_{+,i}) + b$, where $J_A$ is the Jacobian matrix of $A(\tilde{S}_{+,i}, \tilde{I}_{+,i})$. The differentiation to set up $J_A$ is handled automatically.

## 2.2 Advection-Diffusion Equation

The Advection-diffusion equation takes the form

$$
\frac{\partial u}{\partial t} = -\beta \cdot \nabla u + \alpha\Delta u + s,
\tag{3}
$$

with $u = u(x, t) \in \mathbb{R}$ being a measurement of the quantities density and the vector field $\beta = \beta(x, t) \in \mathbb{R}^2$ being the velocity field of e.g. wind over our domain. The diffusion coefficient $\alpha = \alpha(x, t)$ controls the diffusive spread of $u$ and $s = s(x, t)$ is a source term, with $s_+ = s(x, t + h)$.

As boundary condition, we assume Dirichlet boundary conditions $u|_{\partial\Omega} = 0$.

The explicit part, using integration by parts, takes the form:

$$b = \langle \phi, \tilde{u} \rangle + h\theta \langle \phi, \beta \cdot \nabla\tilde{u} \rangle + h\theta\alpha \langle \nabla\phi, \nabla\tilde{u} \rangle - h\theta \langle \phi, s \rangle - h(1 - \theta)\langle \phi, s_+ \rangle$$

and the implicit part

$$A(\tilde{u}_+) = \langle \phi, \tilde{u}_+ \rangle - h(1 - \theta)\langle \phi, \beta \cdot \nabla\tilde{u}_+ \rangle - h(1 - \theta)\alpha \langle \nabla\phi, \nabla\tilde{u}_+ \rangle$$

that is linear in (the parameters of) $\tilde{u}_+$, such that we can solve a linear system

$$A\tilde{u}_+ = b.$$

## 2.3 Wave Equation

The wave equation is a prototype time-dependent PDE, in which we added a damping term to damp simulated waves over time. The wave equation we will use takes the following form

$$\frac{\partial^2 u}{\partial t^2} + b\frac{\partial u}{\partial t} = \Delta u, \quad u(x, 0) = 0. \tag{4}$$

On the boundary of the domain we impose Robin boundary conditions $\alpha u + \beta\eta \cdot \nabla u = 0$, but with a local initial disturbance to start a wave that mimics an initial tsunami-wave. Due to the second derivative in the time domain, the equation doesn't suit our opening definition, but can easily be extended into a suitable system of PDEs:

$$\frac{\partial u}{\partial t} = v, \quad \frac{\partial v}{\partial t} + b\frac{\partial u}{\partial t} = \Delta u.$$

By replacing the time-derivatives, using the trial functions $\tilde{u}, \tilde{v}$ in the first equation, we obtain an implicit and explicit side again

$$(1 - (1 - \theta)^2 h^2\Delta + h(1 - \Theta)b)\tilde{u}_+ = (1 + (1 - \Theta)hb + \theta(1 - \theta)h^2\Delta\tilde{u} + h\tilde{v}$$
$$\tilde{v}_+ = h\Delta((1 - \theta)\tilde{u}_+ + \theta\tilde{u}) + b\tilde{u} - b\tilde{u}_+ + \tilde{v}+,$$

from which the weak formulation can be derived easily. We solve the first equation (linear) for (the parameters of) $\tilde{u}_+$, and receive (the parameters of) $\tilde{v}_+$ straightforward from the second equation.[1]

---

1. The methodology we use here for an undamped ($b = 0$) wave equation, as well as similar numerical code, can be found in this tutorial of the used library https://www.dealii.org/current/doxygen/deal.II/step_23.html. We made only smaller adaptions to the code.

## 3 Dataset Creation

To compute the solutions of the three presented PDEs, we generate a mesh of the domain $\Omega$ with the meshing software *gmsh* (Geuzaine and Remacle, 2009). The implementation of the FEM is done with *deal.ii* (Arndt et al., 2023), which is written in *C++* and abstracted the full implementation around the FEM, but this can be done with any other FEM library.

As an underlying domain $\Omega$ for both the epidemiological PDE and the advection-diffusion PDE, we use the shape of Germany. We evaluate the solutions $\nu(x,t)$ of the PDEs (spatially) on a set of points that have the coordinates of administrative regions in Germany: 400 *NUTS-3* regions. To be more precise we will use the centers of the regions: $V = (x_j)_{j=1,\dots,400} \subset \Omega$. The intuition is the reporting of COVID-19 cases in the same spatial granularity. The FEM allows us the flexible evaluation of $\nu$ everywhere on the domain. The temporal discretization with the Euler method allows the evaluation of $\nu(x,t)$ on all time-points $t = t_0 + ih = ih$ (where $h$ is the Euler stepsize as introduced above), which we shortly refer to as timestep $i$. Thus, for each time-step $i$ we can evaluate $\nu(x,t)$ at every point $x_j$ and assign a feature vector to every such region, leading to a tensor $X_i \in \mathbb{R}^{400 \times d}$. For every location $x_j$ this tensor contains an entry, which is the evaluation of the PDE-solution, i.e. the entry of $X_i$ at position $j$ is $\nu(x_j, ih) \in \mathbb{R}^d$. To equip this data with geographical information, we create a set of edges $E$, containing the edge $(x_i, x_j)$ (and $(x_j, x_i)$ since its undirected), iff the two regions $x_i, x_j$ directly neighbour each other (are adjacent). We further weight these edges with the inverse of the distances of the centers of the regions in metres. More details on this can be found in the appendix A. We thus have created a graph

$$\mathcal{G} = ((V, E, X_i))_{i=1,\dots,N}$$

with $400 = |V|$ nodes, $2088 = |E|$ edges and edge features. The mesh for the FEM computation and the resulting graph can be seen in Figure 1. Note, that through the use of the FEM our method is **not** limited to fixed graphs: since the values of $X_i$ are created by evaluating the solution $\nu(x,t)$ on the points of $V$, no changes to the PDE solver needs to be made, to evaluate the solution on different locations every time-step, e.g. make $V$ and thus $E$ time-dependent and create a graph $((V_i, E_i, X_i))_{i=1,\dots N}$. However, to correspond to real world data, we chose a fix $V$. On our GitHub we provide further explanations on how to increase and change the size of the graph $V$ with minimal effort. Note that no re-computation of $\nu(x,t)$ is needed to create larger graphs, and the evaluation of $\nu$ can easily be parallelized, thus our method is suitable to create also very large graphs.

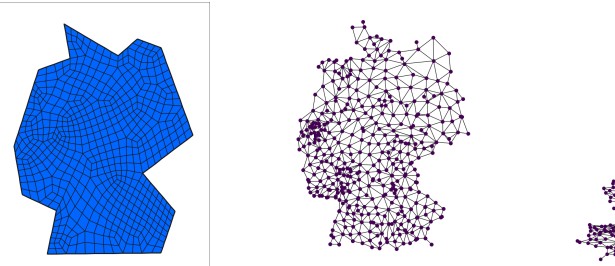

Figure 1: The mesh on the domain for the SI-diffusion equation and the Advection-diffusion equation (left). Independent of this mesh, we build a Graph using administrative regions (*NUTS-3*), their adjacencies, and the distances of the centers (center). Note, that we used the underlying geo-coordinates to plot this graph (center), while the later used and constructed graph has no trivial unique representation (right).

For the wave equation, the domain $\Omega$ is an imaginary (similar to the German) coastline, and the set of points on which we evaluate the solution of the wave equation are chosen randomly. Their respective adjacencies are selected with a Delaunay triangulation, and equipped with inverse distances to form a graph $\mathcal{G}$ with $325 = |V|$ nodes and $1858 = |E|$ edges. The mesh and the resulting graph can be seen in Figure 2.

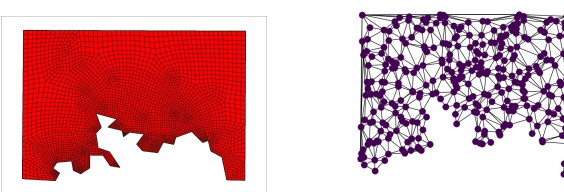

Figure 2: The mesh on the domain for the Wave Equation. On the same domain we generate a set of 325 random points and generate a Delaunay triangulation.

**SI-diffusion equation** We set the parameter $\alpha = 0.22$, the time-weighting parameter $\Theta = 0.5$ effectively resulting in a Crank-Nicolson scheme for accuracy and stability, the Euler stepsize $h_k = 1$ (we fixed this first and only then determined the time-dependent dynamics), and a Newton stepsize $\eta = 0.3$ for stability and convergence. For other PDEs and parameters other choices have to be determined, that cannot be discussed in the scope of this paper, but are subject to the topic of numerical analysis. The underlying domain $\Omega$ can be seen in Figure 1.

We generated 25 scenarios, each with a length of 364 time-steps, resembling an infection scenario over approximately a year. The parameters were chosen heuristically according to their epidemiological interpretation (e.g. disease length around 4.5 days). For every simulation we varied the parameters $r, D$, they can be found in the supplementary material in A and Table 3, leading to a dataset of shape $[9100, 400, 2]$ for [time-steps, nodes, features] in 5:27 hours (wall clock) of computation. Note, that two features exist due to $2 = d = |\{S, I\}|$.

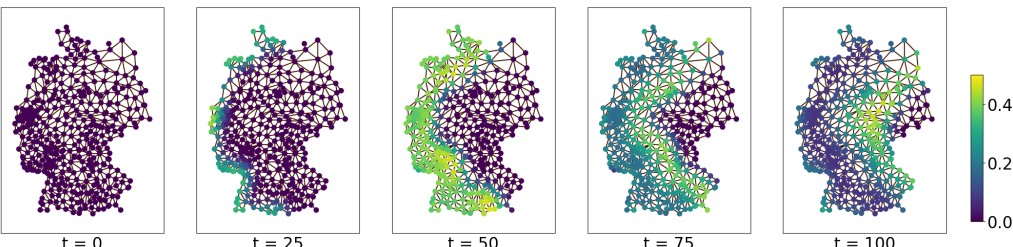

Figure 3: Plot of one scenario of an infection from the diffusion SI-Equation. The Infection wave propagates over the nodes.

**Advection-diffusion equation** For the Advection-diffusion equation, we simulated 54 different parameters right after one another, each with a length of 80 time-steps. Throughout the simulation, we keep $\alpha = -1$ constant and also change $\beta(x,t)$ only every 80 time-steps. For the source term $s(x,t)$ we define a small constant support $\Omega_{s(x,t)} \subset \Omega$ for $s$, such that

$$s(x,t) = \begin{cases} S(t) & x \in \Omega_{s(x,t)} \\ 0 & x \notin \Omega_{s(x,t)} \end{cases}$$

where $\Omega_{s(x,t)}$ is a rectangle within the domain. The choice of $h, T, S(t), \beta(x,t), \Omega_{s(x,t)}$ and further details can be found in the supplementary material A.2 Table 4, resulting in a dataset of shape $[4320, 400, 1]$, for [time-steps, nodes, features] in 0:22h. A spatio-temporal visualization can be found in 4. All selected parameters for this PDE were chosen to generate meaningful and interesting patterns in this visualization method with reasonable sampling rates and frequencies for human eyes and possible ML algorithms.

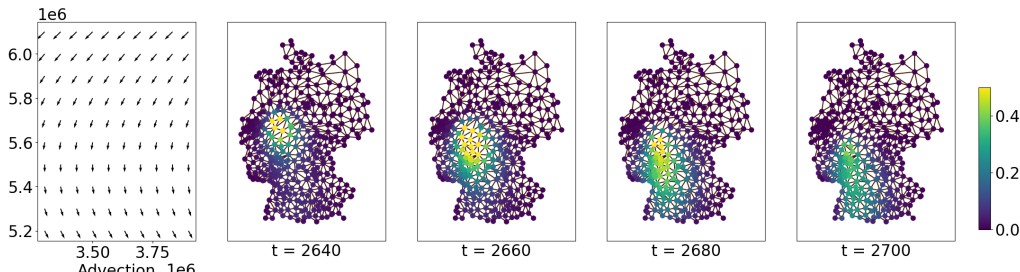

Figure 4: Plot of some advection-field on the left side and an exemplary plot of the solution of the advection-diffusion equation over 4 time-steps.

**Wave equation** For the wave equation, we simulated one consecutive simulation in which we simulated two tsunami waves from different starting points with an initial amplitude of 1. We set $T = 2700000$ and $h = 100000/64$, resulting in a dataset of shape $[1858, 325, 1]$, for [time-steps, nodes, features] in 0:20h. These parameters were again chosen to have reasonable sampling rates on this comparably large domain, and generate visuably confirmable patterns. Details can be found in the appendix A.3.

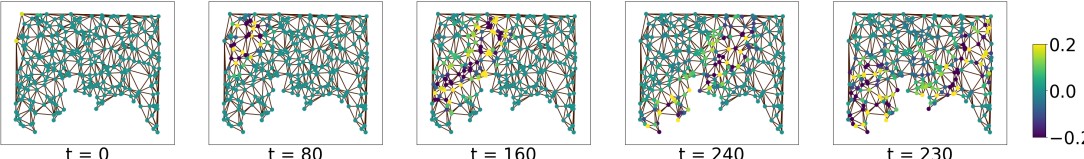

Figure 5: Plot of one scenario of an infection from the Wave-equation.

**Real-world epidemiological datasets** Additionally to the synthetic datasets we created above, we want to compare the epidemiological dataset to real-world data. To do we create three real world datasets: German COVID-19, German Influenza, and Brazilian COVID-19. The numbers of infected people were taken from public sources. The German numbers were connected with the graph $\mathcal{G}$ from the SI-diffusion equation, for the Brazilian data we created a new graph $\mathcal{G}$ analogously by using geographic data of the regions. Details can be found in the appendix, section A.4.

## 4 Application of Synthetic Data

To showcase the utility of our created datasets, we conduct several machine learning experiments in this section. We first define several interesting (spatio-) temporal machine learning architectures in section 4.1. In section 4.2 we define three different benchmarking scenarios, to compare the performance of the previously defined machine learning models. To demonstrate that the usage of our data exceeds pure theoretical studies, we additionally perform experiments of transfer-learning from our synthetic data onto three real-world datasets from epidemiology in section 4.3.

### 4.1 (Spatio-) Temporal Models

We want to test solely data-centric models and do not take any knowledge of the underlying PDE into account, such as its structure or parameters, as done by machine learning-based approaches to solve PDEs (Li et al., 2021; Raissi et al., 2019).

We test both temporal and spatio-temporal models. More specific details can be found in the supplementary material B and the published material online. The presented examples naturally do not take all possible models into account, but offer a good overview. In particular models specialized for traffic data such as GraphWaveNet (Wu et al., 2019) or STG-ODE (Fang et al., 2021) are not taken into account. Besides their specialization in traffic data, they have another drawback as they only operate on a single predefined graph, embedded in the architecture of the models. This makes transfer-learning or generalization onto unseen graphs (or domains) impossible, which will be carried out in section 4.3.

**Repetition** The repetition model is a naive baseline that simply repeats the last given value.

**RNN** Recurrent-Neural Networks based on GRUs (Cho et al., 2014) have proven to be successful for sequence prediction. During training, teacher forcing is applied. During validation forecasts are produced autoregressively.

**TST** Time Series Transformer (TST) is a Transformer-based (Vaswani et al., 2023) architecture designed for temporal forecasting. It utilizes encoder and decoder layers to

capture temporal dependencies, and a positional encoding depending purely on the time-step. During training, teacher forcing is applied. During validation forecasts are produced autoregressively.

**MP-PDE** Given that the underlying data is based on solutions of a PDE, we also test a model based on a Message-Passing PDE-solver (MP-PDE) from Brandstetter et al. (2022), but will not pass any information of the underlying PDE to our model. The model consists of an encoder, a processor, and a decoder. The encoder creates node-wise embeddings of the context data. The processor consists of an MP-GNN, operating on a single graph with embedded features. The decoder is a 1D convolution applied node-wise, and a special update rule, that propagates the last value through the next time-step.

**RNN-GNN-Fusion** Motivated by the PDE itself, we aimed to separate the time from the space dimensions by building an RNN (analogously to the abovementioned model) to encode the underlying ODE and an MP-GNN to emulate the diffusion. RNN and GNN run parallel, their outputs are combined in a convex combination. During training, teacher-forcing is applied. During validation forecasts are produced autoregressively. In the classification of Gao and Ribeiro (2022) this would be classified as *time-and-graph*.

**GraphEncoding** We recreated a model from a recent contribution from the field of ML-based epidemiological forecasting (Nguyen et al., 2023). This model encodes the contextual time-steps separately in a shared GNN. The encoded graphs are then propagated through an LSTM (Hochreiter and Schmidhuber, 1997) network. To achieve a forecast for multiple days, this network is applied autoregressively. In the classification of Gao and Ribeiro (2022) this would be classified as *graph-then-time*.

### 4.2 Epidemiological Benchmarking

To benchmark the presented architectures from section 4.1, we define three tasks on the synthetic epidemiological dataset, created from the SI Eq. 2, that are motivated by forecasting tasks with real-world data. While we wanted to present thorough experiments on this dataset, results with a similar experimental setup, but on the other two datasets from Eq. 3 and Eq. 4, can be found in the appendix D.1.

We will proceed with the epidemiological dataset which we split 76/12/12 along the time axis into a train, test and validation dataset. The splits are multiples of four, since 4% of the dataset are exactly one wave/infectious scenario since we simulated 25 different scenarios, and wanted to prevent data-leakage across scenarios and evaluate on full waves. Further we omit any information about the Susceptible $S$, as this reflects real-world data. On this data, we define the following tasks for machine learning models.

**Forecasting on clean data** The most straightforward task is a simple forecast of the next $n$ timesteps, based on the last $m$ timesteps of inputs. We set $m = n = 14$. The input data into the models therefore are 14 consecutive graphs, sharing the same adjacency, or one graph with 14 node features: $(V, E, X_{i,..,i+13})$. The targets are simply $X_{i+14,..,i+27}$. As a test- and training loss we use the RMSE over all samples, nodes in $V$, and forecasted timesteps $m$.

**Stability: Noise on test data** Since usually there is heavy noise on real-world input data, it is a highly relevant and interesting scenario to test the robustness of the tested machine learning models. We adopt the experimental setting from the previous forecasting

benchmarking, but add noise on the node features $X_{i,...,i+13}$ of the test dataset, but not onto training data. This enables us to study some aspects of the models' robustness in a controlled setting. Further, this reflects real-world data, on which noise exists but is variable due to changed measurements, or delayed reporting. We studied two different types of noise:

- We found the Gaussian noise with distribution $\mathcal{N}(0, 0.01)$ to be an interesting setting for the normalized dataset. A plot of the noisy data can be found in the appendix C Fig 9.

- We also studied noise which reflects the failure of sensors/reports and therefore the occurrence of reported zeros instead of real values. Guided by this scenario, we replace 10% of the test data with zeros to further study the models' robustness.

**Denoising: noise on context data** We seek to study the abilities of different models to implicitly denoise input data, by both training and testing on noisy context data. We use the same Gaussian noise and dropout noise as in the previous experiments.

We executed the benchmarking experiments each three times with different random initializations and trained all models until convergence. Details on the setup and computational aspects can be found in supplementary material B or on our given GitHub. The resulting RMSEs over all samples from the test datasets for the different tasks can be found in Table 1. A visual display of the results can also be found in Fig 10, Fig 11 and Fig 12.

Table 1: **Overview of Performance compared to different tasks.** We take the RMSEs over all samples and all forecasted time-steps from the validation dataset. The $\pm$ indicates the standard deviation out of 3 training runs with random initialization respectively. We scaled all values up by the factor 100 for readability.

| Model | Forecasting | Stability | | Denoising | |
|---|---|---|---|---|---|
| | | Gauss | Dropout | Gauss | Dropout |
| Repetition | $2.43 \pm 0$ | $2.82 \pm 0$ | $3.27 \pm 0.01$ | $2.82 \pm 0$ | $3.27 \pm 0.01$ |
| MP-PDE | $\mathbf{1.08 \pm 0.04}$ | $2.04 \pm 0.1$ | $2.21 \pm 0.1$ | $1.34 \pm 0.01$ | $\mathbf{1.11 \pm 0.09}$ |
| RNN-GNN-Fusion | $1.14 \pm 0.16$ | $\mathbf{1.30 \pm 0.04}$ | $\mathbf{1.70 \pm 0.1}$ | $\mathbf{1.28 \pm 0.07}$ | $1.58 \pm 0.16$ |
| RNN | $1.77 \pm 0.10$ | $1.88 \pm 0.1$ | $2.84 \pm 0.1$ | $1.71 \pm 0.16$ | $3.2 \pm 0.11$ |
| GraphEncoding | $4.40 \pm 1.03$ | $4.47 \pm 1.03$ | $4.86 \pm 0.84$ | $5.67 \pm 3.36$ | $4.59 \pm 0.28$ |
| TST | $1.24 \pm 0.13$ | $2.45 \pm 0.1$ | $2.58 \pm 0.1$ | $2.37 \pm 0.19$ | $2.69 \pm 0.04$ |

The RNN-GNN Fusion model, as well as the MP-PDE, generally exhibits strong performance, effectively leveraging the spatial aspect of the data. However, the performance of GraphEncoding underscores that simply incorporating a GNN does not necessarily boost performance. Therefore, meticulous benchmarking of different architectures on public datasets like ours is critical for the advancement of spatio-temporal graph machine learning.

## 4.3 Transfer Learning to Real-World Data

While the benchmarking of different model architectures already provides valuable insights, we aim to further illustrate the utility of synthetic data in developing foundational fore-

casting models for epidemiology or pre-training models on tasks on which data is sparse. We, therefore, seek to demonstrate whether pre-training on our provided synthetic data can lead to improved performance on real-world tasks. To carry out this experiment we train and evaluate the models from the previous section on three real-world epidemiological datasets: (1) German COVID-19, (2) German Influenza, and (3) Brazilian COVID-19. The prediction task is again a 14-day forecast based on 14 days of input. Simultaneously, we pre-train the same models first on the synthetic dataset based on the SI-diffusion equation, and only then shortly retrain (fine-tune) them on the respective real-world data. We then compare the difference in performance. Note, that the Brazilian data requires an additional knowledge transfer onto an unseen domain (i.e. Graph). Details on the real-world datasets can be found in section A.4. The intuition behind this experiment is to test, whether the epidemiological principles, known to humans and expressed in the form of PDEs (with only few parameters), can be taught to machine learning models to generalize onto much more complex real-world data. More general, this transfer-learning experiment tests if the knowledge, encoded in the PDEs, can be used in data-driven applications through pre-training and transfer learning.

The outcomes of the transfer-learning experiments were highly favorable for pre-training on our dataset, underscoring the effectiveness and possible impact of our proposed method of data synthetization on real-world data and problems. Even with the very limited parameter space of the epidemiological PDE, the models learned epidemiological principles and rules expressed through the PDE, which led to improved performances in the real-world scenario. This underlines the importance and potential to integrate knowledge from the field of modeling with PDEs, to the field of temporal graph machine learning in general. Further, this shows how our method addresses the data scarcity problem in this field and potentially other fields. The results can be found in Table 2. A graphic representation, containing the actual RMSEs and comparing the models against each other can be found in the appendix, see fig. 13.

Table 2: Change in validation-loss of the models in the transfer-learning tasks in percent. A negative number indicates improved performance through our method, i.e. by pre-training on synthetic data as opposed to training solely on the actual data.

| Model | German COVID-19 | German Influenza | Brazilian COVID-19 |
|---|---|---|---|
| MP-PDE | $-7.92\%$ | $6.96\%$ | $-16.88\%$ |
| RNN-GNN-Fusion | $-30.57\%$ | $-19.04\%$ | $-45.58\%$ |
| RNN | $-11.34\%$ | $-16.13\%$ | $-42.61\%$ |
| GraphEncoding | $-8.39\%$ | $-6.91\%$ | $-35.45\%$ |
| TST | $-11.73\%$ | $4.48\%$ | $-12.49\%$ |
| mean | $-13.99\%$ | $-6.13\%$ | $-30.60\%$ |

Only two out of 15 experiments exhibited a slightly decreased performance, while 13 showed an increased performance, frequently to a significant extent. Especially on the Brazilian COVID-19 dataset, some models showed an increase in performance up to 45%.

Especially the previously already favorable RNN-GNN-Fusion model exhibits a significant boost in performance across all datasets.

## 5 Access to code and data

The code and data can be found on github.com/Jostarndt/Synthetic_Datasets_for_Temporal_Graphs.

The code is published under the *GNU LESSER GENERAL PUBLIC LICENSE v2.1.* [2] The created datasets are published under the *CC BY 4.0* license. [3] Both the code and data will be stored on GitHub, which currently allows for indefinite hosting, ensuring long-term availability as long as GitHub remains operational.

## 6 Conclusions

We demonstrated how time-dependent PDEs can be used to create synthetic data for machine learning on graphs. In particular, we have created and published three exemplary datasets that can directly be used by other researchers for further research questions and benchmarking. The described method and code enables researchers to create synthetic temporal graph datasets for individual use cases to support the development of new machine learning methods, in particular, for use cases where data is scarce.

We demonstrate through transfer-learning experiments that pre-training on our synthetic datasets can lead to drastic improvements in real-world data performance. Despite the underlying PDE being relatively limited in flexibility and parameters, these experiments showed a significantly increased performance for some models in our experimental setup. Additionally, as an application of our datasets, we have demonstrated a benchmarking of various models, showcasing that rigorous testing of common architectures was a substantial research gap due to a lack of sufficient data.

Our datasets enables others to build, test, and compare new models for three different applications on large amounts high-quality data, stemming from a fully controllable simulation. We believe this controllability is also advantageous in reducing human biases, since such bias is more likely to stand-out through the compact and explicit representation of PDEs compared to implicit biases in other data-collection processes. However, we want to mention that theoretically our method could still transfer a bias from PDE-based modeling to data-driven applications through a transfer learning task, potentially leading to biased performance even after fine-tuning, although we consider this risk to be very limited.A comparable epidemiological benchmarking with publicly accessible data is unknown to the authors, and especially conducting such a benchmarking with a comparable amount of data was not possible prior to our contribution. We also strongly encourage others to adapt the code and methodology, such as geometry, parameters, scenarios, or PDEs. This not only enables researchers to fulfill individual needs, but also enables others to explore the capacity of machine learning models to simulate specific behaviors of PDEs.

In Summary this paper presents a method for generating individual PDE-based graph datasets, offers three readily available synthetic datasets, provides a benchmarking, and highlights significance through drastic improvements of performance on real-world data

---

2. www.gnu.org/licenses/old-licenses/lgpl-2.1.html.
3. creativecommons.org/licenses/by/4.0/legalcode.en.

in exemplary applications. This work lays the foundation for the development of further datasets and research problems, inviting others to build upon our findings.

**Limitations** The adaptations of our code can become quite complex and more advanced changes can require considerable coding effort and expertise. While exchanging the domain, parameters, sampling rates, or graph structure can be done with little effort, extending or even replacing the PDE can be a very demanding task, requiring a deeper in-depth understanding of the PDE, the FEM, and the employed FEM software. Also our method is constrained to PDEs that can be solved through the FEM, which reaches its computational limits in high-dimensional problems quickly. While our epidemiological experiments demonstrates significant success on real-world data, other applications may need a much greater number of parameters to derive insights for machine learning models, which could also be limited by computational aspects.

## Broader Impact Statement

Spatio-temporal machine learning models and PDEs have diverse applications, including epidemiology, weather forecasting, traffic prediction, and more. Our contribution could be used both for foundational research, as well as pre-training models in domains in which data is sparse.

Our datasets and the creation method leverage PDEs, which are controllable and well-structured. While our work has the potential to enhance data-driven methods for real-world applications and contribute to foundational research, we believe the societal implications are minimal. We aim for a significant improvement in prediction systems, and the authors do not anticipate any negative societal impacts. Our PDEs explicitly do not discriminate against different groups of society or individuals, thus we also see no potential of biased assumptions.

Given the absence of identified drawbacks in our datasets and methodology, we primarily see opportunities for other researchers to expand our datasets, further enhancing their detail and the performance of our PDEs.

## Acknowledgments and Disclosure of Funding

This work was supported by the Federal Ministry for Economic Affairs and Climate Action (BMWK) as grant DAKI-FWS (01MK21009A); the Senate of Berlin and the European Commision's Digital Europe Programme (DIGITAL) as grant TEF-Health (101100700), the European Union's Horizon Europe research and innovation program (EU Horizon Europe) as grant MedEWSa (101121192), and the Fraunhofer Internal Programs as grant PREPARE (40-08394).

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

## Appendix A. Details on: Dataset Creation

The datasets were created with different parameters of the dynamics of the PDEs. We described the key steps to numerically solve the presented equations in the main paper. Further details on the selected parameters to extend section 3 might be valuable to understand the published code and data or to develop more suitable models and will be presented here. The following computations are executed on two AMD EPYC 7543 32-Core Processors.

The evaluation points to create the Graph $\mathcal{G}$ were created by finding the centers of the German *NUTS3* regions. Geographical data regarding the regions are publicly available. [4] The data was processed with the use of *GeoPandas* (Jordahl et al., 2020), *shapely* and *SciPy* (Virtanen et al., 2020) by finding centers of each region, adjacent regions and their distances. Some code regarding this can be found in a notebook on our GitHub under /additional_resources/point_and_mesh_generation.ipynb. The underlying graph of the wave-equation dataset was created with the same geometry files. The evaluation points were chosen randomly, the adjacencies were created using a Delaunay triangulation.

### A.1 SI-Diffusion Equation

The epidemiological dataset from the SI diffusion equation (2) was created by running 25 simulations, each with different parameters, with a length of 364 time-steps. The parameters of the individual simulations can be found in Table 3.

---

4. mis.bkg.bund.de/trefferanzeige?docuuid=D38F5B40-9209-4DC0-82BC-57FB575D29D7, which is published under the dl-de/by-2-0 license (https://www.govdata.de/dl-de/by-2-0 )

Table 3: **Experiment Ids and their respective parameters.** The parameters $r$ and $D$ of the published 25 simulations. "Id." denotes the experiment identifier. The omitted experiment identifier corresponds to different initial values or boundary conditions and can be found in the published code for further simulations. $D$ was scaled in the table by the factor $10^{-8}$ for simplicity of the notation.

| Id. | $r$ | $D \cdot 10^{-8}$ |
|---|---|---|
| 0 | 0.6 | 2 |
| 5 | $1 + (0.2sin(5\mathrm{e}{-}6\pi x_1)sin(5\mathrm{e}{-}6\pi x_2)$ | 2 |
| 10 | $0.7 + (0.6sin(5\mathrm{e}{-}6\pi x_1)sin(5\mathrm{e}{-}6\pi x_2)$ | 2 |
| 15 | $0.5 + (0.3sin(2\mathrm{e}{-}6\pi x_1)sin(5\mathrm{e}{-}6\pi x_2)$ | 2 |
| 20 | $1.1 + (0.1sin(8\mathrm{e}{-}6\pi x_1)sin(5\mathrm{e}{-}6\pi x_2)$ | 2 |
| 25 - 45 | same as 0 - 20 | $(1 + 0.1\ sin(5\mathrm{e}{-}6\pi x_1)sin(5\mathrm{e}{-}6\pi x_2))$ |
| 50 - 70 | same as 0 - 20 | 3 |
| 75 - 95 | same as 0 - 20 | 0.8 |
| 100 - 120 | same as 0 - 20 | $(1 + 0.2sin(5\mathrm{e}{-}6\pi x_1)sin(5\mathrm{e}{-}6\pi x_2))$ |

The resulting simulations were concatenated to a single large dataset. The concatenation is possible since Infected $I$ in the individual converges to zero at the end of every individual simulation. Although the concatenation is not smooth we consider the discontinuity as negligible. A plot of the 25 different scenarios over time can be found in Figure 6.

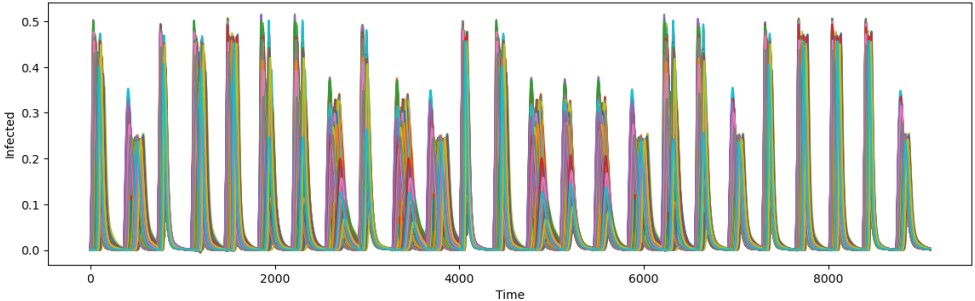

Figure 6: The dataset generated from the SI-diffusion equation over time. There are 400 lines plotted, each representing the values of an individual node over time.

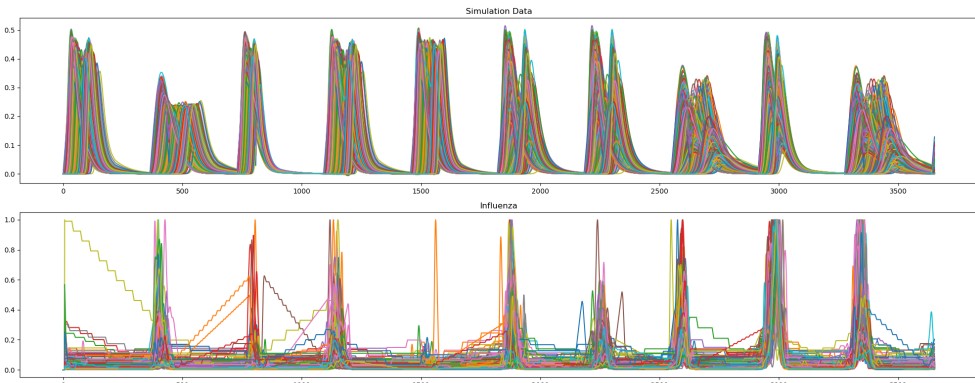

Figure 7: The SI-diffusion dataset(top) compared against the real-world German Influenza dataset(bottom). The step patterns in the Influenza dataset are presumably artefacts of the data collection/management pipeline.

## A.2 Advection-Diffusion equation

The dataset created from the advection-diffusion equation (3) consists of a single, connected, but long simulation. We define 54 different but consecutive sets of dynamics of the equation, each with a length of 80 time-steps. We set $h = \frac{10^9}{8}$, $T = 54 \cdot 10^{10}$.

The support $\Omega_{s(x,t)}$, necessary to describe the source term $s(x,t)$, is a rectangle $[a \cdot 10^6, 1.01a \cdot 10^6] \times [b \cdot 10^6, 1.01b \cdot 10^6]$ on the domain.

The source term $s(x,t)$ was specified in the main paper via $S(t)$ which can be recovered by scaling the temporal dimension $S(t) = \hat{s}(t \cdot 10^{-10}) \cdot 10^{-10}$ from Table 4 as scaled characteristic function, i.e.

$$\mathbb{1}_I(t) = \begin{cases} 1 & t \in I \\ 0 & t \notin I. \end{cases},$$

Where $I \in \mathbb{R}$ is some interval. Also, $\beta(x,t)$, and $a,b$ can be found in Table 4.

Table 4: **Parameters of the advection-diffusion equation** The parameters of the simulation for the advection-diffusion equation."Id." represents 80 consecutive time-steps. The factor $10^4$ in the column defined $\beta$ was used for simplicity of the notation. Note that in the characteristic functions $\mathbb{1}$ also a modulus operation is used, such that after Experiment Id 6, the same dynamics as in 1 starts, i.e. the intervals were shifted to the right. Ids. 7-12 does not match first columns

| Id. | $\hat{s}(t)$ | $\Omega_s$ | $\beta(x) \cdot 10^4$ |
|---|---|---|---|
| 1 | $-32\mathbb{1}_{[0,0.1]} - 22\mathbb{1}_{[0.8,0.9]}$ | $a = 3.45, b = 5.4$ | $(-0.5, 0.5)$ |
| 2 | $-27\mathbb{1}_{[1.3,1.4]}$ | same as 1 | same as 1 |
| 3 | $-32\mathbb{1}_{[2.1,2.2]} - 42\mathbb{1}_{[2.8,2.9]}$ | same as 1 | same as 1 |
| 4 | $-28\mathbb{1}_{[3.3,3.4]}$ | same as 1 | same as 1 |
| 5 | $-38\mathbb{1}_{[4.1,4.2]} - 33\mathbb{1}_{[4.8,4.9]}$ | same as 1 | same as 1 |
| 6 | $-30\mathbb{1}_{[5.3,5.4]}$ | same as 1 | same as 1 |
| 7-12 | same as 1-6 | $a = 3.6, b = 5.6$ | same as 1 |
| 13-18 | same as 1-6 | $a = 3.45, b = 5.8$ | same as 1 |
| 19-36 | same as 1-18 | same as 1-18 | $(0.6, 55 - x_1 10^{-6})$ |
| 37-54 | same as 1-18 | same as 1-18 | $(0.5, 2.5x_1 10^{-6} - 1.375)$ |

Similar to the source term $s$, we also initialize $u$ once on the support $\Omega_{s(x,0)}$ with

$$u = \begin{cases} 1 & x \in \Omega_{s(x,0)} \\ 0 & x \notin \Omega_{s(x,0)}. \end{cases}$$

## A.3 Wave Equation

The wave equation (4) also consists of a single consecutive simulation. There are two waves throughout the simulation, that start at the boundaries. The boundary conditions are usuallyRobin boundary conditions $\alpha u + \beta \eta \cdot \nabla u = 0$ three exceptions: first during the interval $0 < t < 50000$and on the boundary with $x_1 > 6200000, x_0 < 3400000$, second during the interval $1000000 < t < 1100000$, and on the boundary with $x_1 > 6200000, x_0 > 3800000$, and third during the interval $1800000 < t < 1900000$, and on the boundary with $x_1 > 6200000, 3500000 < x_0 < 3600000$.

On these intervals and regions, the Dirichlet boundary $g$ are respectively a constant factor $(0.8, 1, 0.9)$ for the wave's amplitude, that can easily be adapted for further simulations. we set $b = 0.000005$. The first wave can be seen in Figure 5.

## A.4 Creation of Epidemiological Real-World Datasets

The three real-world datasets from section 3 are based from epidemiological data, which can be found online.

1. The Brazilian COVID-19 dataset has 27 nodes and 1093 time-steps spanning 2019-2022 after concatenation and linear interpolation for daily resolution and is publicly accessible. [5]

2. German COVID-19 data can be found at github.com/robert-koch-institut/SARS-CoV-2-Infektionen_in_Deutschland, with 1539 time-steps.

3. German Influenza dataset can be found at survstat.rki.de/ with our curation having 5256 time-steps.

All 3 datasets were smoothed with a 7-day moving average.

The graph describing the spatial connection of the *NUTS3* regions in Germany was created above. The Brazilian geospatial data used for constructing the graph connectivity as described in section A can be accessed at https://www.ibge.gov.br/en/geosciences/territorial-organization/territorial-meshes/18890-municipal-mesh.html.

A visualization can be found in Fig. 8.

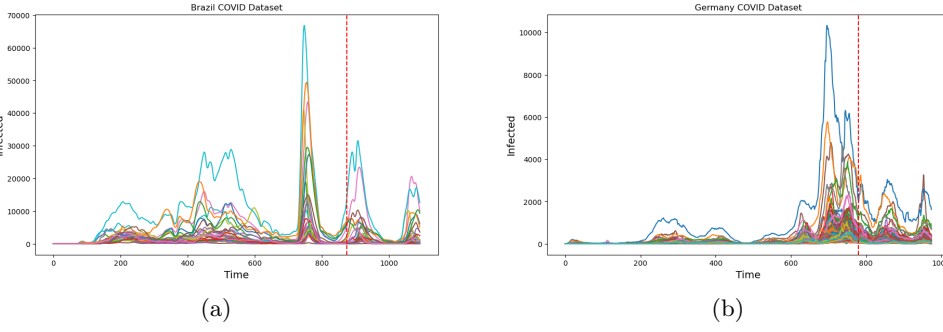

Figure 8: Left: Brazilian COVID dataset. Right: German COVID dataset. In both plots, each curve represents a region, and the vertical red line indicates the training/evaluation cutoff.

These datasets will additionally be released upon acceptance.

## Appendix B. Details on: Applications of Synthetic Data

In our experiments for epidemiological applications from section 4, we trained and evaluated the models on the data based on the SI-diffusion equation in 32-bit float representations. The employed training epochs for the different tasks and models can be found in Table 5.

---

5. sisaps.saude.gov.br/painelsaps/atendimento

Table 5: **Overview of Training Epochs and Computational Aspects for Different Models.** Number of parameters, their respective training times for the forecasting benchmarking, and maximum training epochs for all experiments from section 4 for each model. The exact training times and epochs depend on the specific training run due to random initialization and early stopping.

| Model | # Params | Train Time | #Epochs |
|---|---|---|---|
| MP-PDE | 623k | 57m | 30 |
| RNN-GNN-Fusion | 70k | 2:29h | 30 |
| RNN | 67k | 1:16h | 30 |
| GraphEncoding | 382k | 3:42h | 30 |
| TST | 344k | 1:51h | 50 |

The exact hyperparameters can be found in the implementation on GitHub in the regarding parameter files, exemplary under /ml/mp_pde/mp_pde.yml for the MP-PDE model. The parameter files for other models can be found in their respective directories. The hyperparameters were chosen as optimal for each model experimentally. We trained all models for each task with the Adam optimizer until convergence (early stopping).

## Appendix C. Details on Epidemiological Benchmarking

A visualization of some data with Gaussian noise from the stability and denoising experiments can be seen in Figure 9.

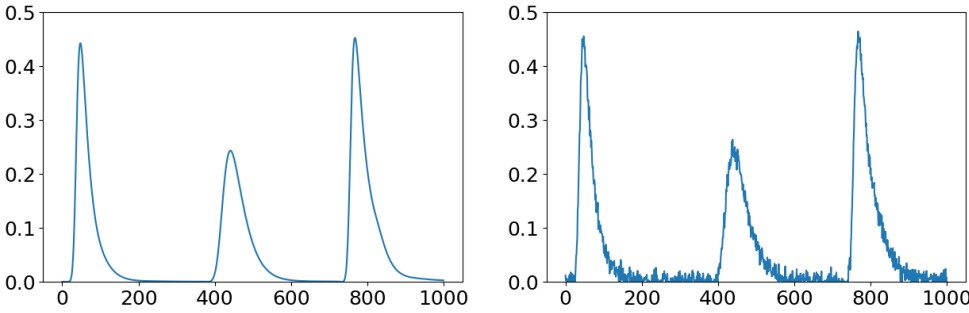

Figure 9: Illustration of the effect of additive noise sampled from the distribution $\mathcal{N}(0, 0.01)$ from task 2 and task 4. On the left is the time series from a random graph node; on the right is its respective noise-added time series. The plotted section is the first 1000 time-steps of the training dataset.

A visual comparison of RMSEs for each time-step in the forecasting experiment for different models can be seen in Fig 10

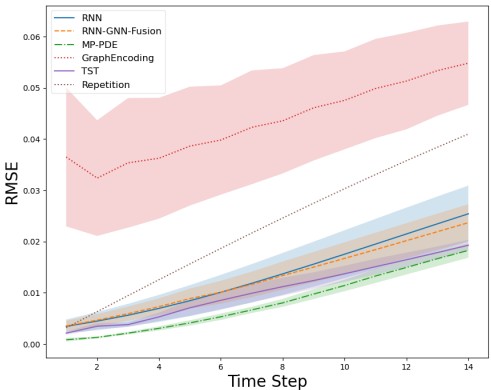

Figure 10: The RMSE over all samples from the validation dataset. The less saturated colors represent the standard deviation for different initializations.

A visual comparison of RMSEs for each time-step in the stability experiments for different models can be seen in Fig 11.

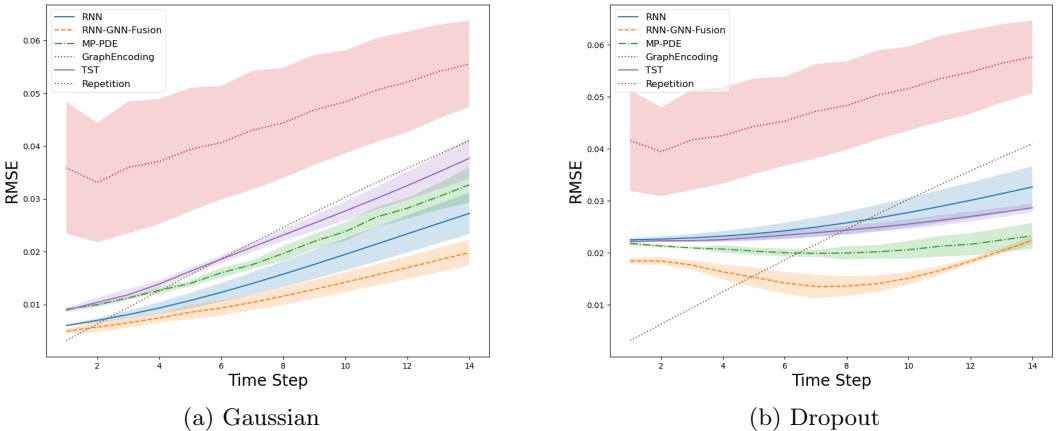

(a) Gaussian

(b) Dropout

Figure 11: Plots of the RMSE of the presented models during the stability experiments for each predicted time-step on the validation dataset. The models were trained on the synthetic dataset, and evaluated on a synthetic dataset with: **a)** additive Gaussian noise, **b)** dropout noise. The less saturated colors represent the standard deviation for different initializations.

A visual comparison of RMSEs for each time-step in the denoising experiments for different models can be seen in Fig 12.

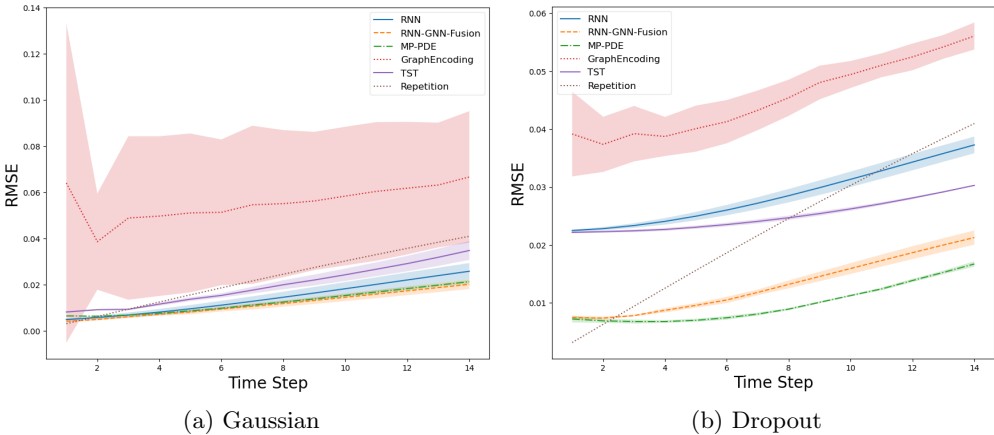

(a) Gaussian                    (b) Dropout

Figure 12: Plots of the RMSE of the presented models during the denoising experiments for each predicted time-step on the validation dataset. The models were trained and evaluated on the synthetic dataset. The models were trained to predict based on context data, with **a)** additive Gaussian noise, **b)** dropout noise. The less saturated colors represent the standard deviation for different initializations.

## Appendix D. Details on Transfer Learning to Real-World Data

To have a better visual comparison between the predictive performances of the models and the degree of their improvement in the transfer learning experiment, we created Fig. 13.

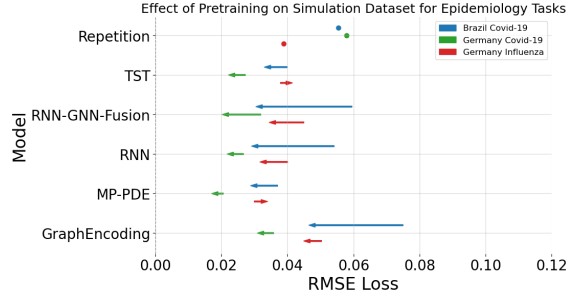

Figure 13: Plots of the RMSEs of the presented models during the transfer-learning experiments. The arrows point from non-pre-trained to pre-trained models.

### D.1 Additional Experiments

We repeated the Forecasting experiment from section 4, more specifically section 4.2 on the two other created datasets, i.e. the Advection-diffusion dataset and the wave-equation dataset.

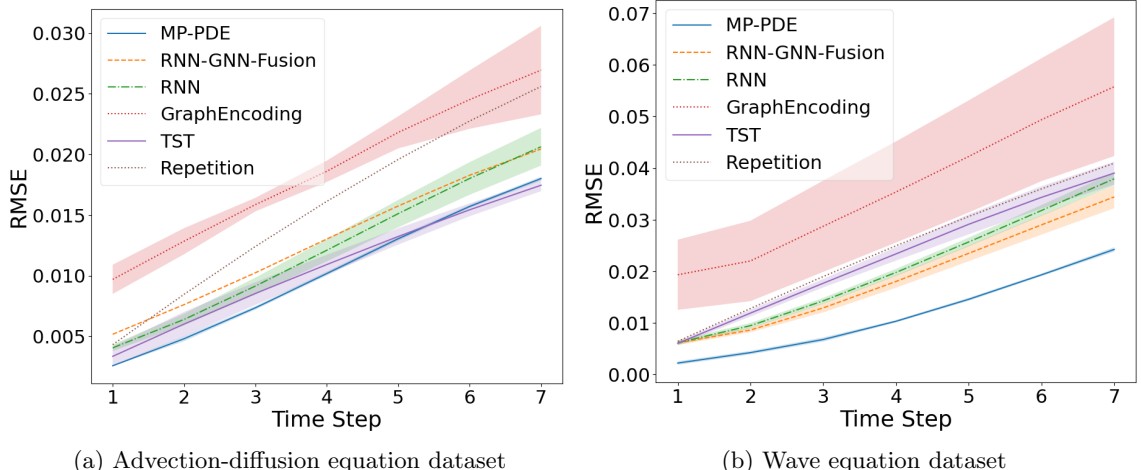

(a) Advection-diffusion equation dataset  (b) Wave equation dataset

Figure 14: **a)** The RMSE for 7-timestep forecasts on the dataset generated by Advection-Diffusion Equation. **b)** The RMSE for 7-timestep forecasts on the dataset generated by Wave Equation.

Table 6: **Performance Comparison Across Models for Different Datasets**

| Model | Advection-Diffusion Equation | Wave Equation |
|---|---|---|
| Repetition | $0.0156 \pm 0$ | $0.0244 \pm 0$ |
| MP-PDE | $\mathbf{0.0115 \pm 0.0001}$ | $\mathbf{0.0139 \pm 0.0003}$ |
| RNN-GNN-Fusion | $0.0134 \pm 0.0009$ | $0.0213 \pm 0.0013$ |
| RNN | $0.0135 \pm 0.0010$ | $0.0234 \pm 0.0008$ |
| GraphEncoding | $0.0195 \pm 0.0012$ | $0.0383 \pm 0.0101$ |
| TST | $0.0117 \pm 0.0006$ | $0.0256 \pm 0.0014$ |

Supplementary Material for DMLR (seperate PDF)

## Appendix E. Our Datasheet

Our datasheet from Gebru et al. (2021) can be found below.

### E.1 Motivation

**For what purpose was the dataset created?** *Was there a specific task in mind? Was there a specific gap that needed to be filled? Please provide a description.*

The Dataset was created to fill a research gap of large high-quality, flexible, easily accessible temporal graph datasets. The first use-case in mind was benchmarking temporal graph machine learning architectures from the field of epidemiology. We then wanted to extend the datasets and showcase the flexibility of our method.

**Who created the dataset (e.g., which team, research group) and on behalf of which entity (e.g., company, institution, organization)?**

All involved people are listed as authors of this paper. The authors are all employees of Fraunhofer Heinrich Hertz Institute, Berlin, Germany.

**Who funded the creation of the dataset?** *If there is an associated grant, please provide the name of the grantor and the grant name and number.*

This work was supported by the German Federal Ministry for Economic Affairs and Climate Action (BMWK) as grant DAKI-FWS (01MK21009A).

**Any other comments?**

No.

### E.2 Composition

**What do the instances that comprise the dataset represent (e.g., documents, photos, people, countries)?** *Are there multiple types of instances (e.g., movies, users, and ratings; people and interactions between them; nodes and edges)? Please provide a description.*

The instances represent measurement values of geographical administrative units (in our case German NUTS3 regions) or a sensor network. Additionally, the published dataset contains adjacencies and inverse distances of these points to form temporal graphs.

**How many instances are there in total (of each type, if appropriate)?** There are three different datasets, the dataset based on the SI-diffusion equation and advection-diffusion equation share the underlying graph, which has 400 nodes and 2088 adjacencies. The dataset based on the wave equation consists of 325 nodes and 1858 adjacencies. The dataset based on the SI-diffusion equation consists of 9100, the dataset based on the advection-diffusion equation consists of 4320 time-steps and the dataset based on the wave equation consists of 960 time-steps.

**Does the dataset contain all possible instances or is it a sample (not necessarily random) of instances from a larger set?** *If the dataset is a sample, then what is the larger set? Is the sample representative of the larger set (e.g., geographic coverage)? If so, please describe how this representativeness was validated/verified. If it is not representative*

*of the larger set, please describe why not (e.g., to cover a more diverse range of instances, because instances were withheld or unavailable).*

This is a synthetic dataset created from PDEs. The points on which we evaluated the PDEs solution are part of the assumptions. Therefore this is not part of a larger dataset. However, the dataset can be extended by additional points in the graph if necessary.

**What data does each instance consist of?** *"Raw" data (e.g., unprocessed text or images) or features? In either case, please provide a description.*

Each value on the dataset (besides the adjacencies) represents a different measurement. In the case of the SI-equation, the measurements describe an infectious density such as cases per capita. The wave equation and advection-diffusion equation also represent some measurement but are rather an abstract density or height, hence we do not give any unit of measurement, as this is a synthetic dataset.

**Is there a label or target associated with each instance?** *If so, please provide a description.*

This is regression data, consisting of temporal graphs. Each value in the dataset corresponds to a value on a graph at a certain time and can be used as a target for regression. Possible applications can be found in Section 4.

**Is any information missing from individual instances?** *If so, please provide a description, explaining why this information is missing (e.g., because it was unavailable). This does not include intentionally removed information, but might include, e.g., redacted text.*

N/A as this is a synthetic dataset.

**Are relationships between individual instances made explicit (e.g., users' movie ratings, social network links)?** *If so, please describe how these relationships are made explicit.*

Yes with the published adjacencies the relationships are made explicit.

**Are there recommended data splits (e.g., training, development/validation, testing)?** *If so, please provide a description of these splits, explaining the rationale behind them.*

No recommended data splits were given.

**Are there any errors, sources of noise, or redundancies in the dataset?** *If so, please provide a description.*

There might be numerical errors in the synthetic datasets stemming from the process of solving PDEs which we consider insignificant, as the FEM with known for its accuracy and the employed spatial resolution was relatively high. In the dataset based on the SI-diffusion equation there are 25 concatenated simulations, between these simulations there are also small errors as the simulations only converge relatively slowly to zero but the following simulations are initialized as zero.

**Is the dataset self-contained, or does it link to or otherwise rely on external resources (e.g., websites, tweets, other datasets)?** *If it links to or relies on external resources, a) are there guarantees that they will exist, and remain constant, over time; b) are there official archival versions of the complete dataset (i.e., including the external*

*resources as they existed at the time the dataset was created); c) are there any restrictions (e.g., licenses, fees) associated with any of the external resources that might apply to a dataset consumer? Please provide descriptions of all external resources and any restrictions associated with them, as well as links or other access points, as appropriate.*

The data is self-contained.

**Does the dataset contain data that might be considered confidential (e.g., data that is protected by legal privilege or by doctor–patient confidentiality, data that includes the content of individuals' non-public communications)?** *If so, please provide a description.*

No.

**Does the dataset contain data that, if viewed directly, might be offensive, insulting, threatening, or might otherwise cause anxiety?** *If so, please describe why.*

No.

If the dataset does not relate to people, you may skip the remaining questions in this section.

### E.3 Collection Process

**How was the data associated with each instance acquired?** *Was the data directly observable (e.g., raw text, movie ratings), reported by subjects (e.g., survey responses), or indirectly inferred/derived from other data (e.g., part-of-speech tags, model-based guesses for age or language)? If the data was reported by subjects or indirectly inferred/derived from other data, was the data validated/verified? If so, please describe how.*

The data was acquired by numerically solving PDEs.

**What mechanisms or procedures were used to collect the data (e.g., hardware apparatuses or sensors, manual human curation, software programs, software APIs)?** *How were these mechanisms or procedures validated?*

The data was collected using software and can be recreated by other users. During the creation, a residual is given which can be seen as validation.

**If the dataset is a sample from a larger set, what was the sampling strategy (e.g., deterministic, probabilistic with specific sampling probabilities)?** The dataset was not a sample from a larger dataset. However, it can be extended, as this is a synthetic dataset by adapting the simulation process.

**Who was involved in the data collection process (e.g., students, crowdworkers, contractors) and how were they compensated (e.g., how much were crowdworkers paid)?** All involved people are listed as authors of this paper and receive compensation as they are employees of Fraunhofer Heinrich Hertz Institute. To list them here again: Jost Arndt, Utku Isil, Michael Detzel, Wojciech Samek, and Jackie Ma.

**Over what timeframe was the data collected?** *Does this timeframe match the creation timeframe of the data associated with the instances (e.g., recent crawl of old news articles)? If not, please describe the timeframe in which the data associated with the instances was created.*

This is a synthetic dataset stemming from numerical solutions of PDEs, the timeframe of their solution is irrelevant to their content. However, the simulations were run in 2024 but can be carried out again within a few hours.

**Were any ethical review processes conducted (e.g., by an institutional review board)?** *If so, please provide a description of these review processes, including the outcomes, as well as a link or other access point to any supporting documentation.*

No.

If the dataset does not relate to people, you may skip the remaining questions in this section.

### E.4 Preprocessing/cleaning/labeling

**Was any preprocessing/cleaning/labeling of the data done (e.g., discretization or bucketing, tokenization, part-of-speech tagging, SIFT feature extraction, removal of instances, processing of miss- ing values)?** *If so, please provide a description. If not, you may skip the remaining questions in this section.*

During the creation of the datasets, every time-step of the simulation resulted in a single file. These files were concatenated to create a single file. Also to create the adjacencies and distances between the points of evaluation, geographical data was processed once. Additionally, no preprocessing, cleaning, or labeling was executed.

**Was the "raw" data saved in addition to the preprocessed/cleaned/labeled data (e.g., to support unanticipated future uses)?** *If so, please provide a link or other access point to the "raw" data.*

The individual files were not saved. The underlying geometries and files can be found online.

**Is the software that was used to preprocess/clean/label the data available?** *If so, please provide a link or other access point.*

The software to concatenate different CSV files or create adjacencies once was not released.

**Any other comments?** No.

### E.5 Uses

**Has the dataset been used for any tasks already?** *If so, please provide a description.*

The epidemiological part of the dataset has been used in the exemplary tasks described in Section 4, which previously has in parts been presented during an ICLR workshop (no proceedings, dataset was not published). The others have not been used.

**Is there a repository that links to any or all papers or systems that use the dataset?** *If so, please provide a link or other access point.*

No, this dataset is new.

**What (other) tasks could the dataset be used for?** The dataset could be used for benchmarking, pretraining, experimenting with noise, experimenting with classification and

many more tasks. However to extend the field of application to other PDEs, our method has to be adapted.

**Is there anything about the composition of the dataset or the way it was collected and preprocessed/cleaned/labeled that might impact future uses?** *For example, is there anything that a dataset consumer might need to know to avoid uses that could result in unfair treatment of individuals or groups (e.g., stereotyping, quality of service issues) or other risks or harms (e.g., legal risks, financial harms)? If so, please provide a description. Is there anything a dataset consumer could do to mitigate these risks or harms?*

This is a synthetic dataset, it is based on the assumptions of the PDE and its parameters.

**Are there tasks for which the dataset should not be used?** *If so, please provide a description.*

No.

**Any other comments?** No.

### E.6 Distribution

**Will the dataset be distributed to third parties outside of the entity (e.g., company, institution, organization) on behalf of which the dataset was created?** *If so, please provide a description.*

This dataset will be made available to the public through this paper.

**How will the dataset will be distributed (e.g., tarball on website, API, GitHub)?** *Does the dataset have a digital object identifier (DOI)?*

The datasets are available as on GitHub.

**When will the dataset be distributed?** Right after the acceptance of this paper.

**Will the dataset be distributed under a copyright or other intel- lectual property (IP) license, and/or under applicable terms of use (ToU)?** *If so, please describe this license and/or ToU, and provide a link or other access point to, or otherwise reproduce, any relevant licensing terms or ToU, as well as any fees associated with these restrictions.*

The data is published under *CC BY 4.0* license, the code it is synthesized under *GNU LESSER GENERAL PUBLIC LICENSE v2.1*.

**Have any third parties imposed IP-based or other restrictions on the data associated with the instances?** *If so, please describe these restrictions, and provide a link or other access point to, or otherwise reproduce, any relevant licensing terms, as well as any fees associated with these restrictions.*

No.

**Do any export controls or other regulatory restrictions apply to the dataset or to individual instances?** *If so, please describe these restrictions, and provide a link or other access point to, or otherwise reproduce, any supporting documentation.*

No.

**Any other comments?** No.

### E.7 Maintenance

**Who will be supporting/hosting/maintaining the dataset?**  The dataset for now is hosted on GitHub which is owned by the Microsoft Corporation.

**How can the owner/curator/manager of the dataset be contacted (e.g., email address)?**  Via email, which is also given on the cover page, and via GitHub.

**Is there an erratum?**  *If so, please provide a link or other access point.*

There is no currently known erratum, besides insignificant numerical errata. If any are discovered at a later point, updated versions of the dataset will be released available on the same platforms.

**Will the dataset be updated (e.g., to correct labeling errors, add new instances, delete instances)?**  *If so, please describe how often, by whom, and how updates will be communicated to dataset consumers (e.g., mailing list, GitHub)?*

There is no plan to regularly update the dataset. However, if flaws are discovered, or additional PDEs are solved, the dataset might be expanded. As this is not planned we cannot give any frequency of updates. The updates can be found on the same platforms as the current datasets.

**If the dataset relates to people, are there applicable limits on the retention of the data associated with the instances (e.g., were the individuals in question told that their data would be retained for a fixed period of time and then deleted)?**  *If so, please describe these limits and explain how they will be enforced.*

N/A

**Will older versions of the dataset continue to be supported/hosted/maintained?**  *If so, please describe how. If not, please describe how its obsolescence will be communicated to dataset consumers.*

The versioning of the datasets will be available on the platforms in their respective versioning systems, e.g. on GitHub via the git version control. The descriptions can also be found on the platforms

**If others want to extend/augment/build on/contribute to the dataset, is there a mechanism for them to do so?**  *If so, please provide a description. Will these contributions be validated/verified? If so, please describe how. If not, why not? Is there a process for communicating/distributing these contributions to dataset consumers? If so, please provide a description.*

Contributions could be made via pull requests on GitHub and have to be reviewed manually. Also, contributions can made manually via email.

**Any other comments?**  No.

## Appendix F. URL to our data and code

github.com/Jostarndt/Synthetic_Datasets_for_Temporal_Graphs

## Appendix G. Author Statement

I, the undersigned author, hereby confirm that I bear full responsibility for any violations of rights, including copyright or intellectual property infringements, that may arise in connection with my submitted work. I affirm that I have obtained all necessary permissions and licenses for any data, images, or materials included in this work. Additionally, I confirm that I comply with the data license and acknowledge the terms under which the data is shared and used.

## Appendix H. Hosting plan:

Both code and data are published on GitHub which is owned by the Microsoft Corporation.

The code is published under the *GNU LESSER GENERAL PUBLIC LICENSE v2.1*. The created datasets are published under the *CC BY 4.0* license.

