# OpenReview forum: "Synthetic Datasets for Machine Learning on Spatio-Temporal Graphs using PDEs"
_DMLR — Accepted by DMLR_

### Review · Reviewer_pw6N · 2025-03-04

**Recommendation:** 3
**Confidence:** 2

**Summary Of Contributions:**

The paper presents a method for generating synthetic spatio-temporal graph datasets based on partial differential equations (PDEs). The authors propose three datasets derived from PDEs modeling epidemiological spread, atmospheric particle movement, and tsunami waves. The paper also benchmarks several machine learning models on the epidemiological dataset and demonstrates the utility of pre-training on synthetic data for improving performance on real-world epidemiological data.

**Strengths:**

See above.

**Audience:**

Yes

**Claims And Evidence:**

Yes

**Datasets And Benchmarks:**

Yes

**Extended Submissions:**

NA

**Requested Changes:**

See above.

**Strengths And Weaknesses:**

Strengths
1. The idea of using PDEs to generate synthetic spatio-temporal graph datasets is innovative and has the potential to address the scarcity of high-quality datasets in this domain.
2. The authors provide the source code and datasets, which could be useful for other researchers to replicate and build upon their work.

Weaknesses
1. The paper does not provide enough technical details on how the PDE solutions are transformed into graph datasets. For example, the process of converting the PDE solutions into graph structures (nodes, edges, and features) is not clearly explained.
2. The experimental results are not convincing. The paper benchmarks several models on the epidemiological dataset but does not compare the performance of these models to state-of-the-art baselines or other synthetic datasets. Additionally, the transfer learning experiments are limited to a single real-world dataset (COVID-19), and the results are not compared to other transfer learning approaches.
3. The paper does not compare the proposed datasets to existing synthetic datasets for spatio-temporal graph learning.
4. While the paper demonstrates transfer learning from synthetic to real-world data, the discussion is limited. The authors do not explore why pre-training on synthetic data improves performance or how this approach compares to other transfer learning techniques.

---

### Review · Reviewer_AgHi · 2025-03-19

**Recommendation:** 3
**Confidence:** 2

**Summary Of Contributions:**

This paper propose a method for generating synthetic dataset based on PDE(partial differential equations) for spatio-temporal graph modeling in ML for different applications. They showcase 3 equations to model different types of disasters and hazards, and bemchmark some ML models on the synthetic dataset to demonstrate their dataset's utility. Also, they released their code for public accessibility.

**Strengths:**

A new way of synthetic dataset generation for PDE-based spatio-temporal graph.
Reproducibility via github
Comparison of different ML architectures.
Experiments shows that the pre-training improved the model performance.
Experiment setting having multiple models and runs.

**Audience:**

Yes

**Claims And Evidence:**

The author is providing some support evidence (experiments, models, datasets) for their claim.

**Datasets And Benchmarks:**

Yes. The author is providing a git for data generation, as well as license. The code is cc-by so i think there won't be too much maintenance

**Extended Submissions:**

N/A

**Limitations:**

It would need sufficient domain knowledge in related area for tuning PDE, are there better way to alleviate this?
The synthetic data, by it's nature, could limit their practicability to real word applications of tasks.

**Requested Changes:**

The weakness part mentioned earlier

**Strengths And Weaknesses:**

Strengths:
A new way of synthetic dataset generation for PDE-based spatio-temporal graph.
Reproducibility via github
Comparison of different ML architectures.

Weakness:
Too much focus on the epidemiological data, where there seems less discussion of the other 2 datasets.
Implementation details - there needs additional discussion on the technical details and underlying reasonings
Additional discussion of potential bias from the synthetic data could be beneficial
Experiments lacks validation set?

---

### Review · Reviewer_4mme · 2025-04-08

**Recommendation:** 3
**Confidence:** 2

**Summary Of Contributions:**

The paper introduces three synthetic spatio-temporal graph datasets generated via PDEs (epidemiology, atmospheric particles, tsunami waves), addressing data scarcity in the field. Key contributions include:
1. First PDE-based spatio-temporal graph datasets, enabling benchmarking and model development.
2. Methodology and code using FEM for customizable dataset generation, supporting domain/parameter adaptations.
3. Publicly available datasets/code, fostering reproducibility and community extension.

**Strengths:**

- Significance: Provides much-needed datasets for a niche area, enabling rigorous model evaluation.
- Prior Work: Builds on PDE and graph ML literature, addressing gaps in dataset availability and adaptability.
- Community Impact: Facilitates foundational research in epidemiology, disaster modeling, and climate science.
- Clarity: Well-structured with detailed methodology, though FEM/PDE sections may challenge non-experts.

**Audience:**

Yes

**Broader Impact Concerns:**

The Broader Impact Statement adequately addresses ethical considerations but could discuss potential misuse (e.g., synthetic data for misinformation) and bias risks if PDE assumptions misalign with real-world dynamics.

**Claims And Evidence:**

Benchmarking (Table 1) and transfer-learning results (Table 2) are rigorous, with RMSE metrics and multiple trials. Wave/particle datasets lack real-world validation; claims about generalizability beyond epidemiology need further evidence.

**Datasets And Benchmarks:**

Yes, there is sufficient information about detailed PDE parameters, code, and GitHub links to ensure reproducibility.

**Extended Submissions:**

Not applicable (no prior version mentioned).

**Limitations:**

1. High-dimensional PDEs may be computationally prohibitive.
2. Real-world validation limited to epidemiology; other use cases remain theoretical.

**Requested Changes:**

1. Clarify dataset accessibility/maintenance plans (hosting, licensing, long-term availability).
2. Discuss computational costs/FEM limitations in depth.

**Strengths And Weaknesses:**

Strengths:
- Addresses critical data scarcity in spatio-temporal graph ML.
- Novel integration of FEM with PDEs for synthetic data generation.
- Comprehensive benchmarking and impactful transfer-learning results.
- High reproducibility via open-source code and datasets.

Weaknesses:
- Limited PDE flexibility: Adapting/new PDEs requires FEM expertise.
- Computational constraints of FEM for high-dimensional problems.